# The potential of novel bacterial isolates from healthy ginseng for the control of ginseng root rot disease (*Fusarium oxysporum*)

**Qiong Li***, **Ning Yan, Xinyue Miao, Yu Zhan, Changbao Chen***

Jilin Ginseng Academy, Changchun University of Chinese Medicine, Changchun, China

* lqginseng@163.com (QL); ccb2021@126.com (CC)

**Data Availability Statement:** All relevant data are within the article.

## Abstract

Ginseng root rot caused by Fusarium oxysporum is serious disease that impacts ginseng production. In the present study, 145 strains of bacteria were isolated from the rhizosphere soil of healthy ginseng plants. Three strains with inhibitory activity against Fusarium oxysporum (accession number AF077393) were identified using the dual culture tests and designated as YN-42(L), YN-43(L), and YN-59(L). Morphological, physiological, biochemical, 16S rRNA gene sequencing and phylogenetic analyses were used to identify the strains as Bacillus subtilis [YN-42(L)] (accession number ON545980), Delftia acidovorans [YN-43(L)] (accession number ON545981), and Bacillus polymyxae [YN-59(L)] (accession number ON545982). All three isolates effectively inhibited the growth of Fusarium oxysporum in vitro and the antagonistic mechanism used by the three strains involved the secretion of multiple bioactive metabolites responsible for the hydrolysis of the fungal cell wall. All three biocontrol bacteria produce indoleacetic acid, which has a beneficial effect on plant growth. From our findings, all three antagonistic strains can be excellent candidates for ginseng root rot caused by the pathogenic fungus Fusarium oxysporum. These bacteria have laid the foundation for the biological control of ginseng root rot and for further research on the field control of ginseng pathogens.

## Introduction

Ginseng (*Panax ginseng* C.A. Mey.) belongs to the family of Wujia, a perennial herb with medicinal and food characteristics and rich nutritional value, mainly produced in China, Korea and other countries [1,2]. In order to ensure the medicinal value of healthy ginseng, a large number of active plant compounds have been isolated and extracted from the root, leaf and fruit parts of ginseng respectively. Specifically, ginsenosides, polypeptides, amino acids and other chemical components have been extracted from ginseng roots, and after their characterization, these compounds were found to possess anti-fatigue [3], anti-tumor [4], anti-diabetic [5], analgesic [6], antioxidant [7] and cardiovascular diseases [8,9].

Ginseng is different from other crops in that it is a crop where the commerciality of the roots is important [10]. It is crucial to cultivate ginseng with healthy roots during the cultivation period of up to 4~5 years [11]. Ginseng is often affected by biotic (microbial pathogens,

**Funding:** This work was supported by the National Natural Science Foundation of China (82204558), Key R & D plan of science and Technology Department of Jilin Province (20220204078YY), the National Natural Science Foundation of China (82073969), the Jilin Provincial Key Science and Technology Project (20200504003YY), the Jilin Provincial Natural Science Foundation Project (YDZJ202101ZYTS015), and the Changchun Science and Technology Development Planning Project (21ZGY13). The funders had no role in study design, data collection and analysis, decision to publish, or preparation of the manuscript.

**Competing interests:** No conflict of interest exits in the submission of this manuscript, and manuscript is approved by all authors for publication. I would like to declare on behalf of my co-authors that the work described was original research that has not been published previously, and not under consideration for publication elsewhere.

pests, etc.) and abiotic (drought, high temperature, low temperature, salinity, etc.) factors during the growth process, resulting in complex disease problems [12]. Ginseng root rot is caused by soil fungal pathogens such as *Fusarium solani* and *Cylindrocarpon destructans*, resulting in dark brown root rot symptoms on the roots [9]. As a member of the genus *Fusarium*, *Fusarium oxysporum* has a wide variety of hosts, strong pathogenicity and serious damage [13,14], and is one of the most destructive pathogenic fungi in the soil, with a high survival rate in the soil under harsh environmental conditions, and in recent years can also cause serious rot on the roots of crops represented by ginseng [15]. According to statistics, the perennial incidence of root rot is around 10% to 20%, which has become a bottleneck problem limiting the sustainable development of ginseng [16].

At present, the main means to prevent fungal diseases of ginseng is to use fungicides, which are effective in controlling plant diseases, but long-term use may cause resistance to the disease and affect soil microorganisms and soil fertility, which cannot effectively alleviate the problem of ginseng diseases [11]. Biological control has become one of the most promising strategies for plant disease control because of its role in protecting the ecological environment, improving human and animal safety, and delaying the resistance of pathogenic bacteria [17,18]. Further, biocontrol products are effective in protecting plants from diseases in an environmentally friendly manner [19]. Current biocontrol products are developed with microorganisms such as bacteria, endophytic fungi and inter-rhizosphere microorganisms as the main raw materials [20–22]. Among them, bacterial biocontrol products are represented by bacteria of the genus Bacillus, which is considered as an ideal microorganism against pathogenic fungi due to its easy colonization of plant surfaces, fast growth rate and resistance [11]. U. R. Radzhabov and K. Davranov have reported that *Bacillus subtilis* SKB 256 is an active antagonist of *Fusarium oxysporum* and *Pseudomonas syringae* [23]. S.N. Sudha et al. have reported that the introduction of the *cry1Ac* Gene of *Bacillus thuringiensis* subspecies into *Bacillus polymyxa* has a dual benefit for rice crops, and they can be used not only as biological insecticide, but also as biofertilizer [24]. It can be said that the use of Bacillus as a biocontrol factor to control plant diseases has become a hot spot for biocontrol research in recent years.

At present, there are few studies to screen biocontrol bacteria to specifically control ginseng root rot. Therefore, this study attempted to isolate biocontrol bacteria with the ability to control ginseng root rot from ginseng inter-root soil that was still growing healthily in heavily cropped ginseng fields. Morphological, physiological, biochemical, 16S rRNA gene sequencing and phylogenetic analyses were used to identify the strains. The morphology of the isolates was observed by scanning electron microscopy (SEM). Optimal growth parameters of the bacterial isolates, regarding temperature, pH, and rotary shaker speed, were also determined. The production and secretion of enzymes that would have a negative impact on *F. oxysporum* were also investigated to provide information on their potential role in the inhibitory activity displayed by the bacterial strains. Lastly, ginseng root disks were used to assess the ability of the bacterial strains to prevent infection by *F. oxysporum* in vitro. The overall purpose of our study was to identify bacterial isolates that exhibited biocontrol activity against ginseng root rot and potentially enhance the growth of ginseng plants. The identified strains provide a foundation for developing biological control of ginseng root rot and other fungal diseases of ginseng (*Panax ginseng*).

## Materials and methods

The experiments involved in this study were carried out in Jilin Ginseng Academy of Changchun University of Chinese medicine, and the work permits of relevant institutions were not involved. It is hereby declared that the situation is true.

## Soil sample collection and isolation of bacteria

Soil samples were collected from a three-year-old *Panax ginseng* plantation located in ZuoJia, JiLin Province, China in June 2021. Six healthy ginseng plants were harvested, and soil attached to the rhizosphere of the plants was collected. All soil samples were placed in aseptic bags and stored at 4˚C prior to processing.

Bacterial isolates were obtained using a serial dilution and plating method. One gram of soil was placed in 99 ml of water in a 250 mL bottle, along with some glass beads. The bottle was then placed at 30˚C on a rotary shaker set at 200 r/min for 40 min [25]. The rhizosphere soil suspension was subjected 5x to a 10-fold dilution and each time, 100ul was spread on petri plates containing Luria-Bertani agar medium (5.0g yeast extract, 10.0g peptone, 10.0g sodium chloride, 15.0g agar, 1.0L distilled water, pH 7.2). The plates were incubated upside down in an incubator at 30˚C until colonies were visible. Single colony isolations were collected, cultured, and maintained for subsequent analysis.

## Screening the bacterial isolates for antifungal activity

*Fusarium oxysporum* (Accession number:AF077393) was obtained from the China General Microbiological Culture Collection Center and cultured on potato dextrose agar (PDA) medium. The inhibitory activity of the bacterial isolates was assessed as the ability to inhibit fungal growth. The antifungal activity of the isolates was determined on PDA using the dual culture tests [26].

In the parallel confrontation test, fungal colonies (6mm) were collected using a cork borer from the edge of a *F. oxysporum* culture exhibiting strong growth and placed on a 90mm diameter petri dish containing fresh PDA. The bacterial isolates were then placed on the PDA medium approximately 2cm away from the fungus [27]. The growth of the fungal colony was observed 7d after inoculation of the plates with the bacterial isolates to assess inhibitory activity, relative to fungal growth on plates without bacterial isolates. The inhibition of fungal growth was calculated using the following equation:

$$Growth\ inhibition\ (\%) = (Dc - Dt)/Dc \times 100\%$$

Where Dc represents control plate colony diameter and Dt represents colony diameter on plates inoculated with the bacterial isolates. treated plate mycelia growth. Each experiment was performed in triplicate and the replicates are presented as the mean ± standard error (n = 3).

## Identification of biocontrol bacteria

**Morphological identification.** The strains YN-42 (L), YN-43 (L) and YN-59 (L) were cultured on LB medium at 25˚C for 3 days. Single bacterial colonies were selected, and colony morphology of each strain was observed and recorded based on Experimental Microbiology 3$^{rd}$ ed. Morphological indicators such as state, color, transparency, and edge were recorded. Gram staining was also performed and observed under a light microscope [27]. Observe the morphology of antagonistic bacteria and measure the size of the bacteria under scanning electron microscopy (SEM). Analyze and measure the size of the bacterium using Image-pro plus 6.0 software.

**Physiological and biochemical characteristics.** Various parameters of the selected bacterial isolates were assessed, based on the Manual for Systematic Identification of Common Bacteria and Bergey's Manual of Determinative Bacteriology [28], including starch hydrolysis, lipid hydrolysis, casein hydrolysis, and gelatin hydrolysis. The following assays were also performed, hydrogen sulfide, catalase, contact enzyme, oxidase, indole, methyl red, citrate

utilization, V–P test, urease, sucrose utilization, phenol red staining, fluorescent pigment, bromocresol purple staining, aerobic growth, and nitrate reduction, as well as several other physiological and biochemical tests.

**Molecular identification and phylogenetic analysis of antagonistic bacteria.** Strains YN-42 (L), YN-43 (L) and YN-59 (L) with strong inhibitory activity against F. oxysporum were preliminarily identified by analysis of 16S rRNA gene sequences. Genomic DNA was isolated using a genomic DNA genome extraction kit (Tian Gen Biochemical Technology Co., LTD., Beijing, China). The 16S rRNA gene was amplified by Polymerase Chain Reaction (PCR) using the universal primers 27F (5′-GATCMTGGCTCAG-3′) and 1492R (5′-TACGGY TACCTTGTTACGACTT-3′). PCR amplification was performed in a thermocycler under the following conditions: initial denaturation at 96˚C for 5 minutes, followed by denaturation at 96˚C for 20 seconds, annealing at 62˚C for 0.5 minutes, extension at 72˚C for 0.5 minutes, and finally extension at 72˚C for 10 minutes. The PCR products were visualized on a 1.0% agarose gel. The PCR products were purified using a magnetic bead purification kit (ShuoMei). Sequencing results were subjected to a BLAST analysis at NCBI. Phylogenetic and molecular evolutionary analyses were performed using MEGA version 6.0 with the following parameters: maximum likelihood using Kimura 2 method with 1000 bootstrap, neighbor joining method and pairwise distance matrix estimated using maximum composite likelihood (MCL) method, and homogeneity rate [29–31].

## Detection of secreted hydrolytic enzymes and secondary metabolites *in vitro*

**Assay of cellulase.** Cellulase activity was assayed according to the method described by Florencio Camila [32]. Cellulase activity was determined by using a plate screening medium containing 1% carboxymethylcellulose. Inoculated plates were incubated at 28˚C, then stained with 0.1% Congo red dye solution for 15 min; the solution was discarded and the cultures were washed with 1 M NaCl for 15 min, and a clear area was observed around the bacteria, indicating cellulase production.

**Detection of chitinase.** Chitinase activity assay was performed according to the method described by Agrawal and Kotasthane [33]. The assay medium was prepared: 4.5 g/L colloidal chitin, 3.0 g/L (NH4)2SO4, 0.3 g/L MgSO4, 2.0 g/L K2HPO4, 1 g/L citric acid monohydrate, 15 g/L agar, 0.15 g/L bromocresol violet, 200 μL Tween 80, pH 4.7, and autoclaved at 121˚C for 15 min and incubated at 28˚C and incubated the biocontrol strain for 72 hours. After incubation, a well-defined area was observed around the bacteria, indicating the production of chitinase.

**Protease assay.** Protease activity was determined by reference to Sokol's method with minor modifications [34]. The biocontrol strain was inoculated into LB agar medium containing 3% skim milk powder and incubated at 28˚C for 72 hours. After incubation, a clear area was observed around the bacteria, indicating protease production.

**Detection of iron carriers.** The CAS assay was applied to detect the production of iron carriers according to the method of Schwyn and Neilands [35]. Specifically, an iron(III) solution was prepared by mixing 1 mM FeCl3 with 10 mL of 10 mM HCl. In another beaker, an orange mixture was prepared by dissolving 60.5 mg of CAS in 50 ml of distilled water, which was then mixed with 10 ml of iron solution to change the color of the solution to purple. Dissolve 72.9 mg of HDTMA (cetyltrimethylammonium) in 40 ml of distilled water, and while stirring, slowly pour the previous purple solution into the HDTMA solution and mix to turn dark blue. pH was adjusted to neutral. After autoclaving (121˚C, 20 min), allow to cool to 50–

60˚C and pour the plate with 900 mL of W A. After inoculation, incubate at 30˚C for 5 d. Observe if a clear orange halo is detected around the bacteria growing in the medium.

**Determination of IAA content.** Cultures of the three selected strains of biocontrol bacteria were incubated in 100 mL YEM broth containing 100ug/ml tryptophan [36]. And the IAA content was determined by the Salkowski's reaction method [37]. Salkowski's reagent containing 0.05 M ferric chloride was dissolved in 1 liter of 35% perchloric acid. An equal volume of Salkowski's reagent was added to 2.0 ml of culture. The contents were mixed by shaking and allowed to stand at room temperature for 30 min to develop a pink color, estimated spectrophotometrically at 500 nm.

**Determination of optimal pH, temperature, and rotary shaker speed requirements for growth of the bacterial isolates.** A single plug of medium (6 mm) containing a colony of each of the bacterial isolates, designated YN-42 (L), YN-43 (L), and YN-59 (L) was placed in LB liquid medium. The growth of the cultures at different temperatures, pH, and at different rotary shaker speeds was assessed. The OD600 value of the culture medium at 600 nm was measured with a UV1102II UV-vis spectrophotometer and growth curves of the bacterial isolates were constructed to determine optimal growth conditions.

Growth at different pH values (pH = 6, 7, 8), temperatures (25˚C, 30˚C, 35˚C), and rotary shaker speeds (160 r/min, 180 r/min, 200 r/min) was assessed.

## Biocontrol efficacy of the bacterial isolates against the ginseng root rot pathogen (*Fusarium oxysporum*)

**Ginseng root collection and disc preparation.** Fresh, healthy, three-year-old ginseng roots ('Damaya') were purchased from the Wanliang ginseng market, Fusong County, Jilin Province, China. The roots were washed with tap water to remove dirt and other material and then immersed in 70% ethanol (EtOH) for 5 minutes, then immersed in 2% sodium hypochlorite (NaClO) for 3 minutes, followed by another immersion in 70% EtOH for 2 minutes, after which the roots were thoroughly rinsed 3x with distilled water. The surface-sterilized ginseng roots were then cut into small discs (approximately $0.5 \times 0.5$ cm) using a sterile surgical knife prior to their use [11].

**Inhibitory assay.** The assay of biocontrol activity was based on the protocol reported by Jang et al. and Song et al [38,39]. The bacterial isolates were cultured in LB liquid medium at 30˚C for 24 hours on a rotary shaker set at 200 r/min. The OD600 of the culture suspension was then adjusted to 0.1. *F. oxysporum*, the ginseng root rot pathogen, was cultured on PDA medium for 7 days. Fungal spores were gently washed from the plates with sterile water and the obtained spore suspension was adjusted to 100 spores/ml [40]. The fungal conidial suspension was then evenly sprayed on the root discs laid out on a tray and left to air dry for 15 minutes. The tray with ginseng root disks in the treatment group were then evenly sprayed with bacterial suspension of one of the isolates. Disks that were not sprayed with a bacterial isolate served as a control. The processed ginseng root disks were then placed in a petri dish, covered with a layer of moist filter paper and cultured at 25˚C for 3 days. All experiments were conducted in triplicate along with one set of control root discs to observe mycelia growth and rot development.

## Statistical analysis

SPSS 21.0 was used for one-way analysis of variance (ANOVA), and significant differences between treatments was determined using a Duncan's new multiple range test ($p < 0.05$). All graphics were produced in Microsoft Excel 2019.

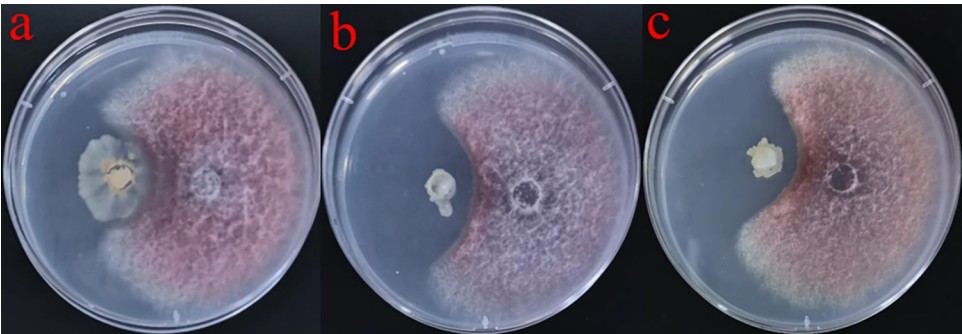

**Fig 1. Antagonistic effect of three biocontrol strains on *Fusarium oxysporum* on plate.**

## Results

### Bacterial isolates obtained from soil samples

In total, 145 isolates were isolated from rhizosphere soil of *Panax ginseng*. Among these, three of the bacterial isolates exhibited inhibitory activity against *Fusarium oxysporum* in vitro. The three strains were then subjected to further analysis to determine their identification and potential as biocontrol agents against ginseng root rot caused by *Fusarium oxysporum*, as well as their potential mechanism of action.

### Determination of inhibitory activity against *F. oxysporum*

The antifungal activity of 145 isolates was screened. The three isolates, designated YN-42(L), YN-43(L) and YN-59(L) had significant inhibitory effect on the growth of *F. oxysporum*. (**Fig 1**). The level of inhibition exhibited by YN-42 (L) against *F. oxysporum* was > 60% (**Table 1**). Repeating the experiments several times indicated that the inhibitory effect was significant. The three strains were then subjected to further analysis and identification.

### Identification of the three bacterial isolates

**Morphology and staining reactions.**    The colony morphology of the three strains was recorded (**Table 2**). The YN-42(L) (*Bacillus subtilis*) colonies on LB medium were white, oval, wrinkled, convex, opaque. The isolate was determined to be gram-positive, and colonies were found to consist of long strips of bacterial cells (0.30 to 0.39 × 0.99 to 1.43 μm) by scanning electron microscopy (**Fig 2A**). (**Fig 2A**). The YN-43(L) (*Delftia acidovorans*) colonies on LB medium were yellow, wrinkled, convex and translucent. The isolate was determined to be gram- negative. By scanning electron microscopy, colonies consisted of short, rod-shaped bacterial cells with a flagellum at one end (0.37 to 0.53 × 1.29 to 2.01 μm) (**Fig 2B**). The YN-59(L) (*Providencia sp.*) colonies on LB medium were white, round, wrinkled, convex, and opaque. The isolate was determined to be gram-positive. SEM observations showed that the cells were rod-shaped (0.68 to 0.72×1.45 to 1.60um) (**Fig 2C**).

**Table 1. Inhibitory activity of biocontrol strains against the mycelial growth of *Fusarium oxysporum*.**

| Strain | YN-42(L) | YN-43(L) | YN-59(L) |
|---|---|---|---|
| Inhibition rate (%) | 68.47±0.006(a) | 62.64±0.007) | 63.74±0.006(b) |

The mean values for the inhibition rate represent preliminary screening measured using the confrontation test.

**Table 2. Morphological characteristics of the bacterial strains isolated from the rhizosphere soil of healthy ginseng plants.**

| | | Strain number | | |
|---|---|---|---|---|
| | | **YN-42(L)** | **YN-43(L)** | **YN-59(L)** |
| Colony morphology | Form | Round | Oval | Round |
| | Color | Faint yellow | Yellow | White |
| | Edge feature | Untidy | Untidy | Untidy |
| | Transparency | Opaque | Translucent | Opaque |
| | Gloss | Glossy | Glossy | Glossy |
| | Texture | Viscous feeling | Viscous feeling | Viscous feeling |
| | Uplift state | Convex | Convex | Convex |
| Individual form | shape | Rod shape | Rod shape | Rod shape |
| | Gram reaction | Positive | Negative | Positive |

**Physiological and biochemical characteristics of the three isolates.** The results of physiological and biochemical assays indicated that the three antagonistic strains all have a strong hydrolytic capacity, including the ability to hydrolyze starch, lipids, and gelatin. While YN-43 (L) could hydrolyze casein, the other two strains did not exhibit this capacity. The IMVIC test indicated that the three strains exhibited the same positive reaction in the indole test and a negative reaction in the other IMVIC assays. Urease and oxidase activity could not be demonstrated in any of the three strains, however, all three strains had the capacity to produce fluorochromes. No ability to reduce nitrate to nitrite was detected and none of the three strains utilized sucrose. The tests also revealed that YN-43 (L) is anaerobic, while the other two strains are partly anaerobic. The phenol red test was negative for all three strains, however, all three strains were positive in the bromocresol violet test (**Table 3**).

**Molecular identification.** DNA was extracted from three bacterial samples and then amplified using PCR method. The bands were clear and bright by gel imaging system. The DNA length of about 1500 bp was consistent with the expected size (**Fig 3**). The PCR products were sent to the company for sequencing and the isolated bacterial strains were identified by sequence analysis and compared using the BLAST analysis tool to confirm 99–100% strain identity coverage. **Fig 4** shows the phylogenetic tree of the three identified bacteria whose accession numbers were separated based on bootstrap values, and the identified bacterial FASTA sequence data publication has been uploaded to NCBI and the respective gene sequence accession numbers were obtained. Phylogenetic trees were constructed in MEGA6. The results showed that YN-42 (L) (accession number ON545980) was most similar to *Bacillus subtilis* (100%), YN-43 (L) (accession number ON545981) was most similar to *Delftia acidovorans* (99%), and YN-59 (L) (accession number ON545982) (100%) was most similar to *Bacillus polymyxa* (100%). All three isolates showed strong inhibition of ginseng root rot. Therefore, we conducted further experiments on them.

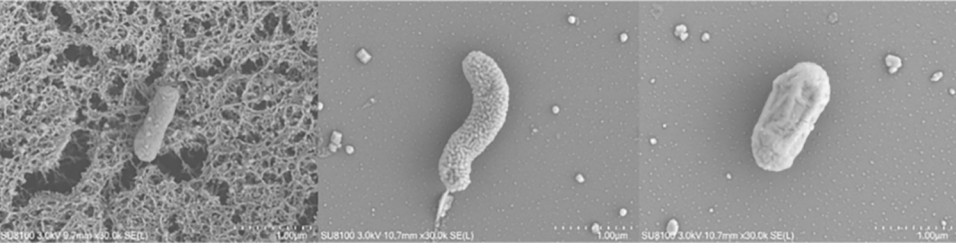

**Fig 2.** Scanning electron micrograph of bacterial cells of (a) YN-42 (L), (b) YN-43(L), and (c) YN-59 (L).

**Table 3. Biochemical and physiological characteristics the three bacterial strains isolated from the rhizosphere soils of healthy ginseng plants.**

| | YN-42(L) | YN-43(L) | YN-59(L) |
|---|---|---|---|
| Amylum hydrolysis test | + | + | + |
| Oil hydrolysis test | + | + | + |
| Casein hydrolysis test | - | + | - |
| Gelatin hydrolysis test | + | + | + |
| Indole reaction | + | + | + |
| Methyl red reaction | - | - | - |
| V-P reaction | - | - | - |
| Utilization of citrate salt reaction | - | - | - |
| Hydrogen sulfide reaction | - | + | - |
| Oxidase reaction | - | - | - |
| Catalase reaction | - | - | - |
| Urease reaction | - | - | - |
| Fluorescence reaction | + | + | + |
| Bromocresol violet staining | + | + | + |
| Phenol red staining | - | - | - |
| Aerobic type | Aerobic | Anaerobic | Facultative anaerobic |
| Nitrate reduction | - | - | - |
| Sucrose utilization reaction | - | - | - |

## Assessment of the mechanism of action in the inhibition of ginseng root rot

Protease [41], cellulase [42], siderophores [43] and chitinase [44] activity was assessed in all three strains in the corresponding assays. An Orange halo was observed around colonies of all three strains cultured on CAS agar medium, indicating siderophore production. All three

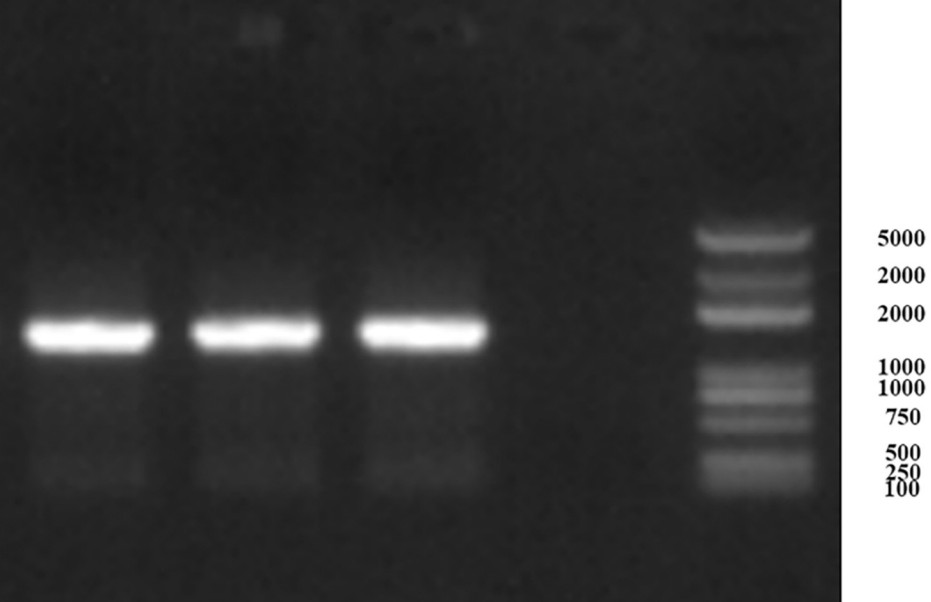

**Fig 3. The results of the PCR amplification reaction system of biocontrol bacteria.**

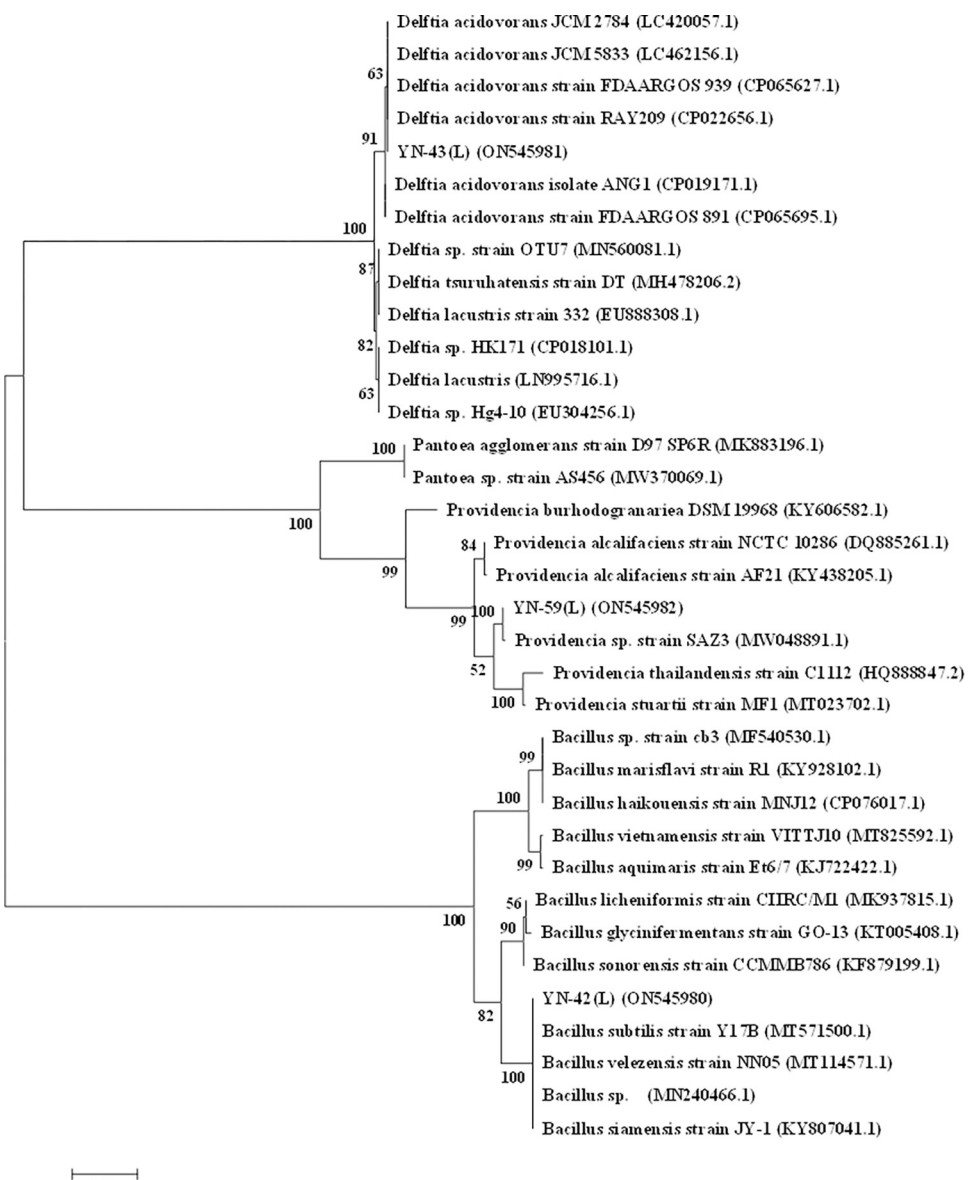

**Fig 4. The phylogenetic tree construction using 16S rDNA gene sequences of the isolated bacterial strains and related strains.** The length of each pair of branches represents the distance between sequence pairs, whereas the units at the bottom of the tree indicate the number of substitution events.

strains exhibited cellulase and protease activity as indicated by the clear area formed around colonies on the CYEA plate containing sodium carboxymethyl cellulose and skim milk (**Table 4**). These results demonstrate that all bacterial strains could secrete siderophores, proteases, and cellulases, however, none exhibited evidence of chitinase activity.

The enzymatic and siderophore properties exhibited by the three strains would contribute to their biocontrol activity against fungal pathogens, such as *F. oxysporum*. Furthermore, a positive pink reaction to the addition of Salkowski's reagent in a test tube containing culture supernatant of the three strains indicated that the YN-42(L), YN-43(L) and YN-59(L) strains, obtained from the rhizosphere soil of *Panax ginseng*, could all produce indoleacetic acid (IAA). The level of IAA produced by the three strains was 5.17mg/L, 5.88mg/L and 24.71mg/

**Table 4. Detection of enzymatic activity in the three bacterial strains isolated from rhizosphere soils of healthy ginseng plants.**

|  | CK | YN-42(L) | YN-43(L) | YN-59(L) |
|---|---|---|---|---|
| Protease detection | - | + | + | + |
| Cellulase assay | - | + | + | + |
| Siderophores detection | - | + | + | + |
| Chitinase assay |  | - | - | - |
| IAA content(mg / L) | - | 5.17b | 5.88b | 24.71a |

L, respectively. These results suggest that in addition to biocontrol activity against *F. oxysporum*, the three isolates could also promote the growth of ginseng plants.

## Optimal culture conditions for the three isolates

The three strains quickly entered the logarithmic growth period at the initial stage of growth when cultured in LB medium. Growth began to level off after 12 hours but the population continued to increase. The cultures entered into stationary phase after 40 hours (**Fig 5A**). The growth rate of the three strains was tested at 25, 30, and 35˚C. While some slight differences were observed, results indicated that 35˚C was optimum for all three strains (**Fig 5B**). Growth of the three strains at different pH values (6.0, 7.0 and 8.0), was also assessed. Results indicated that YN-42 (L) grew best at pH 6.0, YN-43 (L) grew best at pH 7.0, and YN-59 (L) grew best at pH 8.0 (**Fig 5C**). The impact of different speed settings (150, 180 and 200 r/min) on the rotary shaker during culture was also assessed. While some slight differences were observed, in general all three isolates grew best at 200 r/min (**Fig 5D**).

## Biocontrol activity of the three bacterial strains against ginseng root rot evaluated on root disks

Cut sections of ginseng root were first inoculated with the biocontrol strains followed by inoculation with the ginseng root rot pathogen *F. oxysporum*. Results of the *in vitro* test are present

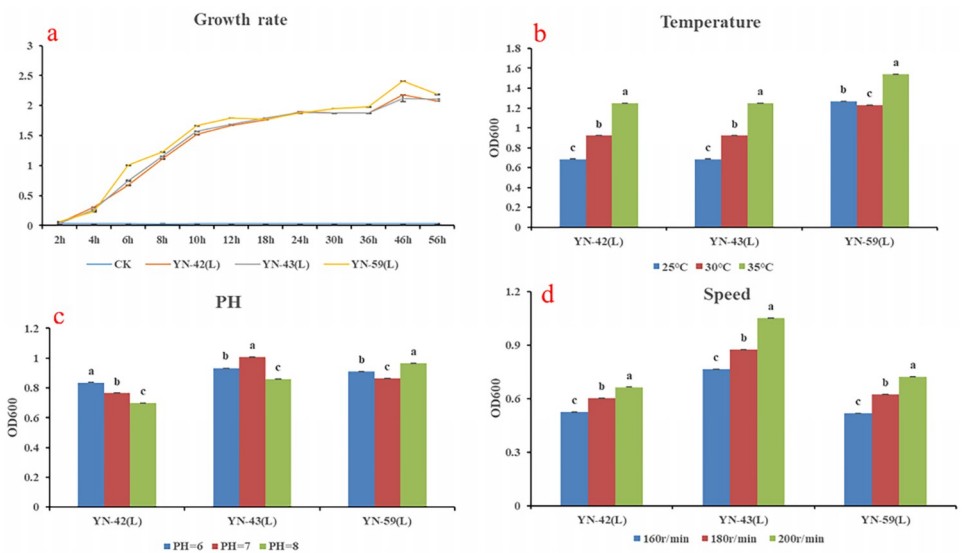

**Fig 5. Growth of the bacterial strains under different temperatures, pH, and rotary shaker speeds.**

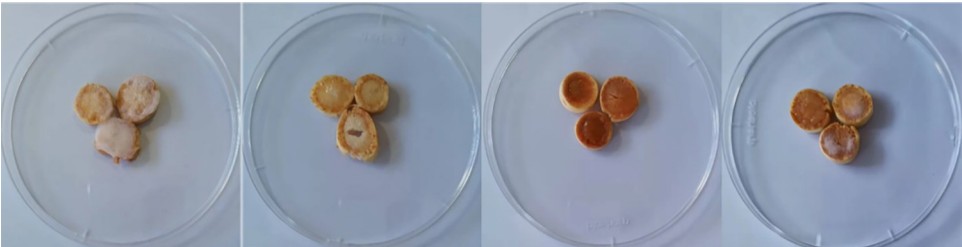

**Fig 6. Ginseng root rot symptoms on ginseng root disks and biocontrol activity of the three isolated bacterial strains.** (a) control (disks not inoculated with the bacterial isolates) exhibited extensive mycelial growth of *F. oxysporum*. (b) Inoculated with YN-42(L) (no mycelial growth). (c) Inoculated with YN-43(L) (no mycelial growth). (d) Inoculated with YN-59(L) (little mycelial growth).

in **Fig 6**. Root disks in the control (not treated with the any of the bacterial strains) all exhibited signs of infection and were covered with mycelia from *F. oxysporum*. In contrast, root disks treated with the bacterial strains exhibited distinct evidence of inhibitory biocontrol activity. All three bacterial strains significantly inhibited the growth of *F. oxysporum*, however, the inhibitory activity exhibited by YN-42 (L) and YN-43 (L) was greater than YN-59 (L).

## Discussion

In plant disease and microbial ecosystems, when plant disease is severe, pathogenic bacteria occupy a favorable position and antagonistic bacteria are in a suppressed state [45]. When effective antagonistic bacteria are able to inhibit the growth and reproduction of pathogenic bacteria, the plant is free from disease or mild disease [46]. Studies have shown that antagonistic bacteria can not only reduce the number of pathogenic bacteria by rapidly occupying relevant physical and ecological sites, but also compete for nutrients and inhibit the growth of pathogenic bacteria, thus slowing down the invasion of pathogenic bacteria [47]. Therefore, screening of fungi, bacteria, actinomycetes and other microorganisms with inhibitory effects on pathogenic bacteria, and the development and application of new biological fungicides on this basis are necessary for the biological control of soil-borne diseases in agricultural fields and even the development of green agriculture [48,49]. In this experiment, 145 strains of bacteria were isolated and purified from the inter-rhizosphere soil of ginseng. Three different biocontrol bacteria, namely YN-42(L), YN-43(L) and YN-59(L), were screened by plate standoff. The results showed that all three antagonistic bacteria showed significant inhibition against *F. oxysporum*, which causes root rot of ginseng. After physiological and biochemical experiments and molecular sequencing, it was found that they showed high identity with *Bacillus subtilis*, *Delftia acidovorans* and *Bacillus polymyxa*, respectively. In addition, this paper reconfirmed that all three strains showed good biological control effects through root plate rewiring experiments, which have certain application value in biological control. At present, there are few studies on the screening of ginseng root rot pathogenic bacteria, and this paper provides a reference for the screening and control of ginseng root rot bacteria.

Antagonistic biotrophic bacteria usually exert their biological control through their metabolites. The three strains screened in vitro were exposed to different metabolites such as indoleacetic acid, iron carriers and essential hydrolases. Indoleacetic acid is an essential secondary plant metabolite that is synthesized by bacteria to promote natural plant growth. In this study, three growth-preventive bacteria YN-42(L), YN-46(L), and YN-59(L) screened from ginseng inter-rhizosphere soil were able to produce indoleacetic acid, indicating that they not only antagonize *F. oxysporum* but also promote plant growth. Kotoky R also reported that Bacillus subtilis SR1 exhibited indole -3- acetic acid (IAA) production ability [50]. As an important

growth hormone for plant growth, indoleacetic acid enhances the vitality of the root system to absorb more nutrients, thus increasing the resistance of the plant itself and improving its immunity to foreign disease invasion [51].

Iron carrier acts as a high-affinity iron chelator [52,53]. When it binds to iron, the organism secretes a soluble iron complex, which is not only essential for the survival of certain bacteria in an iron-limited environment, but is also one of the mechanisms by which biocontrol bacteria inhibit the growth of plant pathogens [54,55].The ability of pathogenic microorganisms to acquire iron from their hosts is one of the key steps in their development in their hosts, Colin Ratledge and Lynn G Dover reported [56]. However, in pathogenic microorganisms, iron carriers can steal iron from host proteins that can then be used by bacteria and other organisms [57]. In this experiment, the three strains screened produced iron carriers by forming an orange halo zone on the ferritin medium, suggesting that the mechanism of antagonism of the three biocontrol strains against root rot pathogens is also related to iron carrier production.

Hydrolases that decompose periplasmic pathogenic microorganisms can be important predictors of plant diseases. Different species of Phytophthora can secrete different cell wall lytic enzymes such as protease, cellulase, and chitinase that exhibit degradation. In the present study, all three strains were positive for amylase, protease and cellulase production. These positive results may reveal that when ginseng root rot pathogenic bacteria infest the plants, the three strains of the biocontrol bacteria defend against them by secreting several of these enzymes to lyse the cell wall of the pathogenic bacteria [58].

## Conclusions

In our study, the dual culture tests and in vitro biocontrol assays were used to identify and confirm the inhibiting effect of three bacterial strains, isolated from the rhizosphere of healthy ginseng plants, against *F. oxysporum*. The study provides a foundation for developing biological control of ginseng root rot disease. The initial tests conducted in this study were all conducted in vitro, and further studies conducted under filed conditions will be required to further validate the efficacy of these bacterial biocontrol agents against ginseng root rot and other soil-borne diseases of ginseng, as well as their ability to promote the growth of ginseng plants.

## Author Contributions

**Formal analysis:** Yu Zhan.

**Funding acquisition:** Changbao Chen.

**Investigation:** Xinyue Miao.

**Writing – original draft:** Ning Yan.

**Writing – review & editing:** Qiong Li.

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
