## [Decision Letter · Decision Letter 0]

5 Aug 2022

PONE-D-22-17252Screening and identification of bacteria with biocontrol activity against ginseng root rot disease and their potential mechanism of actionPLOS ONE

Dear Dr. Li,

Thank you for submitting your manuscript to PLOS ONE. After careful consideration, we feel that it has merit but does not fully meet PLOS ONE’s publication criteria as it currently stands. Therefore, we invite you to submit a revised version of the manuscript that addresses the points raised during the review process.

We look forward to receiving your revised manuscript.

Kind regards,

Rashid Nazir

Academic Editor

PLOS ONE

Journal Requirements:

   "This work was financially supported by grants from National Natural Science Foundation of China (82073969), Jilin Province Major Science and Technology Special Project (20200504003YY), Jilin Province Natural Science Foundation Project (YDZJ202101ZYTS015), and Changchun Science and Technology Development Plan Project (21ZGY13). "

 "This work was financially supported by grants from National Natural Science Foundation of China (82073969), Jilin Province Major Science and Technology Special Project (20200504003YY), Jilin Province Natural Science Foundation Project (YDZJ202101ZYTS015), and Changchun Science and Technology Development Plan Project (21ZGY13). These grants were received by Professor Changbao Chen and played an important role in deciding to publish and prepare manuscripts"

   "No conflict of interest exits in the submission of this manuscript, and manuscript is approved by all authors for publication. I would like to declare on behalf of my co-authors that the work described was original research that has not been published previously, and not under consideration for publication elsewhere."

7. We note you have included a table to which you do not refer in the text of your manuscript. Please ensure that you refer to Table 4 in your text; if accepted, production will need this reference to link the reader to the Table.

Additional Editor Comments:

Please make the changes as per reviewers' comments and prepare the revised manuscript according to journal guidelines.

Reviewers' comments:

Reviewer's Responses to Questions

**Comments to the Author**

1. Is the manuscript technically sound, and do the data support the conclusions?

Reviewer #1: Yes

Reviewer #2: Yes

Reviewer #3: No

2. Has the statistical analysis been performed appropriately and rigorously? 

Reviewer #1: Yes

Reviewer #2: Yes

Reviewer #3: Yes

3. Have the authors made all data underlying the findings in their manuscript fully available?

Reviewer #1: Yes

Reviewer #2: Yes

Reviewer #3: No

4. Is the manuscript presented in an intelligible fashion and written in standard English?

Reviewer #1: Yes

Reviewer #2: Yes

Reviewer #3: Yes

5. Review Comments to the Author

Reviewer #1: The reviewed manuscript entitled "Screening and identification of bacteria with biocontrol activity against ginseng root rot disease and their potential mechanism of action." is generally clear and well written.

The study dealt with a promising scientific approach to inhibiting pathogens through plant growth-promoting rhizobacteria (PGPR). The authors were interested in studying the properties of growth-promoting bacteria and their effect on inhibiting the in vitro growth of pathogenic fungi.

However, the authors have not studied the effect of the bacteria in the pot experiment. It is important to complete the research by studying the effect of growth-stimulating bacteria in the in vivo trails rather than studying it on section cut of the Root.

In general, the research dealt with an important topic and reached promising results that could be applied in the field as an alternative to chemical pesticides. However, I have the following comments:

1. The novelty of the study needs to be highlighted compared to other similar studies.

2. Introduction part: Must contain the whole background regarding the targeted problem and how to solve that problem with a comparison with the literature review; please check and revise accordingly.

3. The discussion seems to be poor; it didn't give good explanations of the results obtained. I think that it must be really improved. Where possible, please discuss the potential mechanisms behind your observations. You should also expand the links with prior publications in the area but try to be careful not to over-reach. For the latter, you should highlight potential areas for future study.

Specific Comments:

Line 1-3: The title could be shortened, e.g. "The potential of novel bacterial isolates from healthy ginseng for the control of ginseng root rot disease (Fusarium oxysporum)".

Keywords: The selected keywords should not be mentioned in the title; please change them accordingly

Section 3.1 : No reference was cited on the different protocols you worked with; how is that? The section must contain recent and related references with more details to be beneficial to broad scientific readers.

Line 109: Spell out the abbreviation LB.

Line 115 : did the pathogen you worked with is identified? If yes, please provide the accession number of the sequence in the GeneBank. If not, the pathogen needs to be identified.

Line 140: Enter the full name "PCR" in the subsection heading.

In the section Phylogenetic Analysis of Antagonistic Bacteria:

Line 148- 151: Several modifications should be done to the text; you should add the information about the different software used for the treatment of sequences (Alignments, assembly.

About the phylogeny analysis and tree construction, you need to know that Neighbor-joining is a clustering algorithm that can make quick trees but is not the most reliable; the method is not acceptable for publication. You need to reconstruct your tree based on the Maximum likelihood using Kimura 2 method with 1000 bootstrap.

In the results section: 4.3 Taxonomic identification of the three bacterial isolates

The different bacterial isolates sequenced in the present study should be published in the GeneBank; the similarity is not enough; you should give accession numbers to your sequence.

The tables are represented by '+’ or ‘-‘, only the yes or no to each of these assays. However, each bacterial isolate will have a difference in the production of these lytic enzymes/ metabolites. This will be evident from the zones of inhibition. Include this data in the table to get a complete visual assessment of this aspect. This is included in the text for a few bacteria in the amylase, protease activity etc. Adding this to the table will give a clearer idea to the reader.

Another important aspect you should verify is the detection of lipopeptides by PCR; while you are working with only Three isolates, you should verify the different genes that may be involved in the potential of different bacteria.

In Figure 1: the phylogenetic tree should contain all the bacterial isolates with their references; not every isolate with its own tree.

In table 4, for IAA content: add the statistical analysis

It is necessary to detail the title of figure 3 and add the standard deviation for Fig.3a

Reviewer #2: The manuscript entitled “Screening and identification of bacteria with biocontrol activity against ginseng root rot disease and their potential mechanism of action” by Li. et al. deals with laboratory observation of bacterial isolates that exhibited biocontrol activity against ginseng root rot and potentially enhance the growth of ginseng plant. The manuscript contains precious laboratory data. However, the reviewer could not find any new insights of the finding due to many articles relating to the recent work were published already, except if the manuscript contains field data of efficacy validation of these bacterial against plant root rot and other soil borne diseases. Herewith, some suggestions.

Line (L) 80-90: Please only emphasis why this research important, rather than explain the result of the research.

L135. Please provide the accession number of the sequences YN-42, YN-43, and YN-59 and do re-construction of phylogenetic analysis by adding the accession numbers (Figure 1.).

In Figure 2, please provide the actual size (….micro meter?) of bacterial cells in SEM.

In table 3, Please rewrite the ghree bacterial strains…… to the three bacterial strains…….

Reviewer #3: The study sought to find possible biocontrol microorganisms for Ginseng root rot disease. The work is essential because it will give an alternate method of reducing foot rot diseases to chemical treatment.

(1) The writing must be thoroughly reviewed for grammatical and spelling errors.

(2) Method 3.6 -Detection of secreted hydrolytic enzymes and secondary metabolites requires additional explanation.

(3) The findings were not supported by evidence such as phylogenetic diagrams, gel photos, inhibitory test images, and so on.

6. PLOS authors have the option to publish the peer review history of their article (what does this mean?). If published, this will include your full peer review and any attached files.

Reviewer #1: **Yes: **Nabil Radouane

Reviewer #2: **Yes: **Oslan Jumadi, Ph.D., Prof.

Reviewer #3: No

---

## [Author Response · Author response to Decision Letter 0]

14 Sep 2022

Editor

1.Please ensure that your manuscript meets PLOS ONE's style requirements, including those for file naming.

Response: We are grateful for your valuable suggestions and we have already reformatted the manuscript and renamed the files according to your suggestion and the PLOS ONE style requirements.

Response: We appreciate your valuable suggestions. The research method used in this study does not involve the work permit of relevant institutions, for which we will provide a statement in the cover letter.

3. This work was financially supported by grants from National Natural Science Foundation of China (82073969), Jilin Province Major Science and Technology Special Project (20200504003YY), Jilin Province Natural Science Foundation Project (YDZJ202101ZYTS015), and Changchun Science and Technology Development Plan Project (21ZGY13). "We note that you have provided funding information that is not currently declared in your Funding Statement. However, funding information should not appear in the Acknowledgments section or other areas of your manuscript. We will only publish funding information present in the Funding Statement section of the online submission form. Please remove any funding-related text from the manuscript and let us know how you would like to update your Funding Statement. Currently, your Funding Statement reads as follows: "This work was financially supported by grants from National Natural Science Foundation of China (82073969), Jilin Province Major Science and Technology Special Project (20200504003YY), Jilin Province Natural Science Foundation Project (YDZJ202101ZYTS015), and Changchun Science and Technology Development Plan Project (21ZGY13). These grants were received by Professor Changbao Chen and played an important role in deciding to publish and prepare manuscripts" Please include your amended statements within your cover letter; we will change the online submission form on your behalf.

Response: We appreciate your valuable suggestions. First of all, we have deleted any text related to funds from the manuscript according to your suggestion, and decided to republish the funds information in the funds statement section of the online submission form according to your proposal. Thank you very much for your suggestion. You can decide to change the online submission form on our behalf. At the same time, we have added two more capital projects to the original ones. We will provide a Funding Statement in the cover letter according to your suggestions.

4. Thank you for stating the following in your Competing Interests section: "No conflict of interest exits in the submission of this manuscript, and manuscript is approved by all authors for publication. I would like to declare on behalf of my co-authors that the work described was original research that has not been published previously, and not under consideration for publication elsewhere." Please complete your Competing Interests on the online submission form to state any Competing Interests. If you have no competing interests, please state "The authors have declared that no competing interests exist.", as detailed online in our guide for authors at http://journals.plos.org/plosone/s/submit-now. This information should be included in your cover letter; we will change the online submission form on your behalf.

Response: Thank you for your suggestions. We will provide a no competing interests statement in the cover letter according to your suggestions.

Response: However, for this problem, we still intend to provide repository information for the data upon receipt, and thank you again for your suggestions

6. PLOS requires an ORCID ID for the corresponding author in Editorial Manager on papers submitted after December 6th, 2016. Please ensure that you have an ORCID ID and that it is validated in Editorial Manager. To do this, go to ‘Update my Information’ (in the upper left-hand corner of the main menu), and click on the Fetch/Validate link next to the ORCID field. This will take you to the ORCID site and allow you to create a new ID or authenticate a pre-existing iD in Editorial Manager. Please see the following video for instructions on linking an ORCID iD to your Editorial Manager account: https://www.youtube.com/watch?v=_xcclfuvtxQ

Response: Thank you for your suggestion. Our corresponding author has obtained an ORCID according to the method you provided. Thank you again.

7. We note you have included a table to which you do not refer in the text of your manuscript. Please ensure that you refer to Table 4 in your text; if accepted, production will need this reference to link the reader to the Table.

Response: We extremely grateful to your visionary comments. According to your comments, we have explained the contents in Table 4 in the relevant parts of the newly submitted version.

Response: We extremely grateful to your visionary comments. However, we still intend to provide repository information and supporting information files for the data when receiving this article, and thank you again for your suggestions.

.

Reviewer 1

1.The novelty of the study needs to be highlighted compared to other similar studies.

Response: We are grateful for your valuable suggestions. In order to emphasize the novelty of this study, we have made extensive adjustments on the contents of the paper in the previous draft. In our resubmitted manuscript, the abstract and introduction were supplemented and modified to reflect the novelty of this study.

2. Introduction part: Must contain the whole background regarding the targeted problem and how to solve that problem with a comparison with the literature review; please check and revise accordingly.

Response: we are very grateful for your valuable suggestion, which made us realize that the introduction is missing the target problem and the overall background of how to solve it. In response, we have reorganized the introductory section and added comparisons to the references cited, which are highlighted in red.

3. The discussion seems to be poor; it didn't give good explanations of the results obtained. I think that it must be really improved. Where possible, please discuss the potential mechanisms behind your observations. You should also expand the links with prior publications in the area but try to be careful not to over-reach. For the latter, you should highlight potential areas for future study.

Response: We extremely grateful to your visionary comments. According to your comments, we have added potential biological control mechanisms in the discussion section, and correspondingly expanded the relevant contents of previous publications in this field according to your suggestions, so as to make the discussion more complete. Specific changes are highlighted in red.

4. Line 1-3: The title could be shortened, e.g. "The potential of novel bacterial isolates from healthy ginseng for the control of ginseng root rot disease (Fusarium oxysporum). Potential of new bacterial isolates from healthy ginseng for the control of root rot (Fusarium acuminatum) in ginseng.

Response: We sincerely thank your professional comments. According to your suggestion, we have revised the title of the manuscript and marked the revised portions in red.

5.Keywords: The selected keywords should not be mentioned in the title; please change them accordingly 

Response: We sincerely thank your professional comments. According to your suggestion, we have changed the keyword and marked the revised portions in red.

6. Section 3.1: No reference was cited on the different protocols you worked with; how is that? The section must contain recent and related references with more details to be beneficial to broad scientific readers.

Response: We sincerely thank your professional comments and careful checks of our study. According to your suggestion, we have corrected the manuscript and marked the revised portions in red.

7. Line 109: Spell out the abbreviation LB

Response: We sincerely thank your professional comments and careful checks of our study. We have spelled the full name of LB completely and the revision details are highlighted by using red colored text.

8.Line 115: did the pathogen you worked with is identified? If yes, please provide the accession number of the sequence in the GeneBank. If not, the pathogen needs to be identified.”

Response: Thanks for your valuable suggestions, the pathogens in the article have been confirmed, and the registration number of the specific Genebank sequence has been supplemented in the relevant positions in the article, and the specific registration number has been marked in red.

9.Line 140 of the specific suggestion: enter the full name of PCR in the section title.

Response: We sincerely thank your professional comments and careful checks of our study. We have spelled the full name of PCR completely and the revision details are highlighted by using red colored text.

10.Line 148- 151: Several modifications should be done to the text; you should add the information about the different software used for the treatment of sequences (Alignments, assembly. About the phylogeny analysis and tree construction, you need to know that Neighbor-joining is a clustering algorithm that can make quick trees but is not the most reliable; the method is not acceptable for publication. You need to reconstruct your tree based on the Maximum likelihood using Kimura 2 method with 1000 bootstrap.

Response: We extremely grateful to your visionary comments. According to your comments, we have used the maximum likelihood method and related methods to construct phylogenetic trees based on your comments, and have added relevant information on constructing phylogenetic trees in the molecular biology identification section, with specific changes highlighted in red.

11. The different bacterial isolates sequenced in the present study should be published in the GeneBank; the similarity is not enough; you should give accession numbers to your sequence 

Response: We sincerely thank your professional comments. According to your comments, we have published the sequenced different bacterial isolates in the GeneBank, and supplemented the sequence accession numbers of the strains in the newly submitted version.

12. The tables are represented by ‘+’ or ‘-’, only the yes or no to each of these assays. However, each bacterial isolate will have a difference in the production of these lytic enzymes/ metabolites. This will be evident from the zones of inhibition. Include this data in the table to get a complete visual assessment of this aspect. This is included in the text for a few bacteria in the amylase, protease activity etc. Adding this to the table will give a clearer idea to the reader. 

Response: We extremely grateful to your visionary comments. According to your comments, we have summarized the hydrolase activity data of the three biocontrol strains in Table 4, and the specific content has been marked in red.

13. Another important aspect you should verify is the detection of lipopeptides by PCR; while you are working with only Three isolates, you should verify the different genes that may be involved in the potential of different bacteria.

Response: Thank you for your valuable comments. Your suggestion is very helpful to us. We know that supplementing this part of the experiment will improve the level of this article as a whole. However, we think that this article has revealed an antagonistic mechanism by detecting the bacteriolytic effect of biocontrol bacteria on pathogenic fungi. The potential relationship between different genes and different bacteria will be reflected in our subsequent research.

14. In Figure 1: the phylogenetic tree should contain all the bacterial isolates with their references; not every isolate with its own tree.

Response: We are very grateful to you for your valuable suggestion. We have adopted the maximum likelihood method to build a phylogenetic tree containing three kinds of biocontrol bacteria according to your suggestion. See Figure 1 in the new version for specific modifications.

15. IAA content in Table 4: Adding statistical analysis.

Response: We are sorry for our negligence of the content of adding statistical analysis to IAA in the former manuscript. According to your suggestion, we have added the statistical analysis of IAA content of three biocontrol strains in Table 4.

16. It is necessary to detail the title of figure 3 and add the standard deviation for Fig.3a.

Response: We sincerely thank your professional comments. According to your comments, we have added the title of Figure 3 and the standard deviation of Figure 3a in a timely manner.

17. The authors have not studied the role of bacteria in pot experiments.

Response: We sincerely appreciate the valuable comments and we fully agree with the importance of this suggestion. However, considering that ginseng has a very long growth cycle, we can’t get effective experimental data if we supplement in vivo validation experiments in a short time. Therefore, we have not supplemented this part of the experiment.

Reviewer 2

1. Line (L) 80-90: Please only emphasis why this research important, rather than explain the result of the research.

Response: Thank you for your constructive comments. We have adjusted this paragraph according to your suggestions and emphasized the importance of this study.

2. L135. Please provide the accession number of the sequences YN-42, YN-43, and YN-59 and do re-construction of phylogenetic analysis by adding the accession numbers (Figure 1.).

Response: Thanks for your valuable suggestions According to your suggestions, we have added the gene sequence registration numbers of the three biocontrol bacteria to the phylogenetic tree in the new version.

3 In Figure 2, please provide the actual size (….micro meter?) of bacterial cells in SEM.

Response: Thank you for your reminding. We have supplemented the actual sizes of the three biocontrol bacteria in our resubmitted manuscript.

4 In table 3, Please rewrite the ghree bacterial strains…… to the three bacterial strains……. ”

Response: We are sorry for our negligence of the spelling errors in the former manuscript. According to your suggestion, we have corrected the spelling errors and the revision details are highlighted by a red font. 

Reviewer 3

1. The writing must be thoroughly reviewed for grammatical and spelling errors Response: We are sorry for our negligence of the spelling errors in the former manuscript. According to your suggestion, we have corrected the spelling errors and the revision details are highlighted by a red font.

2. Method 3.6-Detection of secreted hydrolytic enzymes and secondary metabolites requires additional explanation.”

Response: We are very grateful to you for your valuable suggestion. Under your suggestion, we especially supplemented the methods for detecting hydrolases and secondary metabolites one by one. The specific supplementary contents are reflected in the resubmitted version.

‎3. The findings were not supported by evidence such as phylogenetic diagrams, gel photos, inhibitory test images, and so on.

Response: Thank you for your constructive suggestion. At your suggestion, we timely supplemented the phylogenetic diagrams, the gel photo and the inhibition test image to verify the results of this study.

---

## [Decision Letter · Decision Letter 1]

17 Oct 2022

PONE-D-22-17252R1The potential of novel bacterial isolates from healthy ginseng for the control of ginseng root rot disease (Fusarium oxysporum)PLOS ONE

Dear Dr. Li,

Thank you for submitting your manuscript to PLOS ONE. After careful consideration, we feel that it has merit but does not fully meet PLOS ONE’s publication criteria as it currently stands. Therefore, we invite you to submit a revised version of the manuscript that addresses the points raised during the review process.

We look forward to receiving your revised manuscript.

Kind regards,

Estibaliz Sansinenea

Academic Editor

PLOS ONE

Journal Requirements:

Additional Editor Comments:

The reviewers have commented about the Ms and reccommended minor revision

Reviewers' comments:

Reviewer's Responses to Questions

**Comments to the Author**

1. If the authors have adequately addressed your comments raised in a previous round of review and you feel that this manuscript is now acceptable for publication, you may indicate that here to bypass the “Comments to the Author” section, enter your conflict of interest statement in the “Confidential to Editor” section, and submit your "Accept" recommendation.

Reviewer #1: All comments have been addressed

Reviewer #2: (No Response)

2. Is the manuscript technically sound, and do the data support the conclusions?

Reviewer #1: Yes

Reviewer #2: Yes

3. Has the statistical analysis been performed appropriately and rigorously? 

Reviewer #1: Yes

Reviewer #2: Yes

4. Have the authors made all data underlying the findings in their manuscript fully available?

Reviewer #1: Yes

Reviewer #2: Yes

5. Is the manuscript presented in an intelligible fashion and written in standard English?

Reviewer #1: Yes

Reviewer #2: (No Response)

6. Review Comments to the Author

Reviewer #1: the paper need to address the following comments before publishing

In section 3.2 (Gene bank sequence registration number AF077393.1), change it to

Accession number: AF077393

Change Similarity to Identity in all the text

Revise the sections on molecular and morphological identification sections numbering is missing

Reviewer #2: The manuscript was re-write well based on suggestion of reviewer, but reviewer think that the authors should be mention the accession number in the abstract and the methods.

7. PLOS authors have the option to publish the peer review history of their article (what does this mean?). If published, this will include your full peer review and any attached files.

Reviewer #1: **Yes: **RADOUANE Nabil

Reviewer #2: No

---

## [Author Response · Author response to Decision Letter 1]

21 Oct 2022

Editor：

Response: We sincerely thank your professional comments and careful checks of our study. At your suggestion, to avoid citing papers that have been retracted, we have re-cited the relevant literature, as indicated by the red markings in the references.

Reviewer #1：

In section 3.2 (Gene bank sequence registration number AF077393.1), change it to

Accession number: AF077393

Response: We are grateful for your valuable suggestions. In our resubmitted manuscript, (Gene bank sequence registration number AF077393.1) has been changed to Access number: AF077393 according to your suggestion.

Change Similarity to Identity in all the text

Response: We sincerely thank your professional comments and careful checks of our study. According to your suggestion, we have corrected the manuscript and marked the revised portions in red.

Revise the sections on molecular and morphological identification sections numbering is missing

Response: Thanks for your constructive suggestions. We have added the numbering on molecular and morphological identification sections and marked the revised portions in red.

Reviewer #2: 

The manuscript was re-write well based on suggestion of reviewer, but reviewer think that the authors should be mention the accession number in the abstract and the methods.

Response: We extremely grateful to your visionary comments. According to your comments, we have added the accession number in the abstract and the methods.

---

## [Editor Report · Decision Letter 2]

24 Oct 2022

The potential of novel bacterial isolates from healthy ginseng for the control of ginseng root rot disease (Fusarium oxysporum)

PONE-D-22-17252R2

Dear Dr. Li,

We’re pleased to inform you that your manuscript has been judged scientifically suitable for publication and will be formally accepted for publication once it meets all outstanding technical requirements.

Kind regards,

Estibaliz Sansinenea

Academic Editor

PLOS ONE

Additional Editor Comments (optional):

The authors have incorporated all recommendations made by the reviewers therefore the MS can be accepted in the current form
---

## [Editor Report · Acceptance letter]

1 Nov 2022

PONE-D-22-17252R2 

The potential of novel bacterial isolates from healthy ginseng for the control of ginseng root rot disease (*Fusarium oxysporum*) 

Dear Dr. Li:

I'm pleased to inform you that your manuscript has been deemed suitable for publication in PLOS ONE. Congratulations! Your manuscript is now with our production department. 

Kind regards, 

on behalf of

Dr. Estibaliz Sansinenea 

Academic Editor

PLOS ONE